# Gender Differences and Optimizing Women's Experiences: An Exploratory Study of Visual Behavior While Viewing Urban Park Landscapes in Tokyo, Japan

Ruochen Ma [ID], Yuxin Luo and Katsunori Furuya *[ID]

Graduate School of Horticulture, Chiba University, Chiba 271-8510, Japan
* Correspondence: k.furuya@faculty.chiba-u.jp

**Abstract:** Improving the inclusiveness of urban green spaces and enabling various groups to equally enjoy their benefits is the basis of sustainable urban development. Urban park design generally starts from a gender-neutral perspective, ignoring differences in needs related to gender, particularly women's sensitivity to the environment. This study focused on visual perception and explored gender differences and proposed causes of visual-behavior differences while viewing landscapes. We used photo data from Mizumoto Park in Tokyo and recruited 16 master's students living nearby to participate in an eye-tracking experiment. The results indicate that men and women have different eye-movement patterns and that elemental ratios affect eye movement behavior more among women than men. Moreover, this study found that men gaze longer at trees and more briefly at shrubs, flowers, and artificial elements than women. Attention-grabbing paths/grounds had a negative effect on the aesthetic evaluation of the scene among women but not men. Based on these findings, suggestions for optimizing women's experiences at the visual level are proposed for aspects of vegetation density, visual focus, and road design. This study informs park design and improvement with the premise that gender alters the perception of these environments.

**Keywords:** landscape; gender; urban park; eye-tracking; visual perception; park management

## 1. Introduction

Urban parks provide a range of benefits for residents of all ages and contribute to the sustainability of the urban landscape [1,2]. Using urban green spaces like urban parks can improve the physical and mental health of urban residents, as well as relieve anxiety and stress [3,4]. Understanding the needs and usage of park users is also critical in the development and management of urban parks [5]. The diversity of people matters for park usage [5]. For example, scholars have found that urban park visitors may have vastly different experiences based on gender [6]. Multiple studies have demonstrated that women visit parks less frequently than men and stay for shorter periods, especially in large-scale urban parks with scattered management personnel and large areas of natural environment [7–9]. Women tend to take care of children in urban parks and, unlike men, infrequently participate in activities like exercising alone or with peers [10,11]. However, women exhibit a greater demand for urban green space and emphasize the recreational, health, and social benefits provided by urban parks more than men [12–14]. Richardson and Mitchell (2010) [15] suggested that the quality of green spaces may affect their utilization by women over men. Previous research suggests perceived quality is a significant predictor of green space visit frequency [16]. Therefore, it is necessary to improve the quality of green spaces through design and management to give women a better experience and promote women's visits to urban parks. Most urban park designs use a gender-neutral perspective, ignoring differences in sensitivities and expectations particular to gender [17]. Daniel (2013) [18] argues that such a "neutral" lens perpetuates the inequality and invisibility of women. A separate consideration of feelings based on gender could allow visitors to benefit

from urban parks more equally. Hence, it is important to recognize gender differences in the perception of park landscapes.

Most studies on gender differences in landscape perception used questionnaires. While some studies demonstrate that the difference is small or insignificant [19,20], others report significant differences in perceived safety, naturalness, and aesthetic value [6,21,22]. In addition to subjectively expressed feelings, gender differences can be compared by measuring physiological responses, such as heart rate, blood pressure, skin conductance, and eye movement [23]. Innate biological differences between men and women may lead to different responses to the environment, and social differences and environmental perception may be reflected by physiological indicators [24].

As more than 80% of human sensory input is visual [25], landscape perception and preference are largely dependent on vision [26]. Moreover, providing visual enjoyment is an important function of urban parks. Therefore, this study attempted to explore objective gender differences from a visual perspective, assuming that different visual behaviors result in different experiences between men and women. Wu et al. (2021) [27] pointed out that eye-movement indicators are likely to be predictors of landscape preference. A study by Li et al. (2020) [28] revealed the importance of landscape elements decomposition for visual perception and evaluation. Therefore, we focus on visual behavior in terms of eye-movement patterns and element gaze distribution. We hypothesized that men and women would display different eye movement patterns when confronted with the same scene, that they may focus on different landscape elements and read different information. We proposed that the type and proportion of these landscape elements may affect their evaluation of the scene. Eye-tracking technology provided effective data to test this hypothesis. Eye trackers can accurately record respondents' eye-movement trajectories when observing a scene and generate large amounts of data to analyze observation patterns and focus [27,29]. Multiple studies have used eye tracking to investigate group differences in landscape perception, often examining variables such as expertise, gender, and cultural background [30–33]. Guo et al. (2021) [32] and Wang et al. (2021) [33] noted gender differences in visual preferences for landscape types. Their studies have confirmed the feasibility of eye tracking as a research method; however, gender was included as only one of the demographic variables in studying specific landscape types. To our knowledge, relatively detailed studies of the interrelationships between gender, landscape elements, and visual behaviors are lacking in this field. Therefore, based on these two studies, we conducted a study in Japan. Several studies have reported that women have ambivalent feelings, including interest and fear, when exposed to large natural areas in highly urbanized environments [34,35]. Men generally show no such concerns and can enjoy these areas more freely [35]. Therefore, a large-scale, natural city park was considered more likely to show gender differences in visual behavior than small residential parks. Tokyo, Japan has the largest metropolitan area in the world. Therefore, the largest natural city park in Tokyo was selected as the study site. This study posed the following questions:

- Do eye movement patterns when viewing the park landscape differ between men and women? Are eye movement patterns related to landscape element proportions?
- Do gaze distributions toward various elements when viewing park landscapes differ between men and women? Do gaze distributions affect aesthetic evaluation?

This study used quantitative data analysis to answer the posed research questions. In addition, this study investigated the reasons underlying gender differences based on previous literature.

## 2. Materials and Methods

### 2.1. Study Site

Mizumoto Park in Tokyo was the study site (Figure 1). This metropolitan park opened in 1965 with a planned area of approximately 145 hectares [36]. Although located in the urban area, the original natural resource and historic water bodies are preserved. The park contains waterscapes, water-resistant trees, and rare aquatic plants. Wild birds can be ob-

served all year in the bird sanctuary [37]. The natural environment and abundant facilities provide a variety of scene choices for material sampling of eye-movement experiments. Moreover, the park plays a pivotal role in Tokyo's disaster prevention and helps form Tokyo's water and green framework [38]. Furthermore, this urban park meets the recreational needs of residents in the three jurisdictions—Tokyo, Saitama, and Chiba—which is relatively rare globally. The park had approximately 2.3 million visitors in 2021 [36].

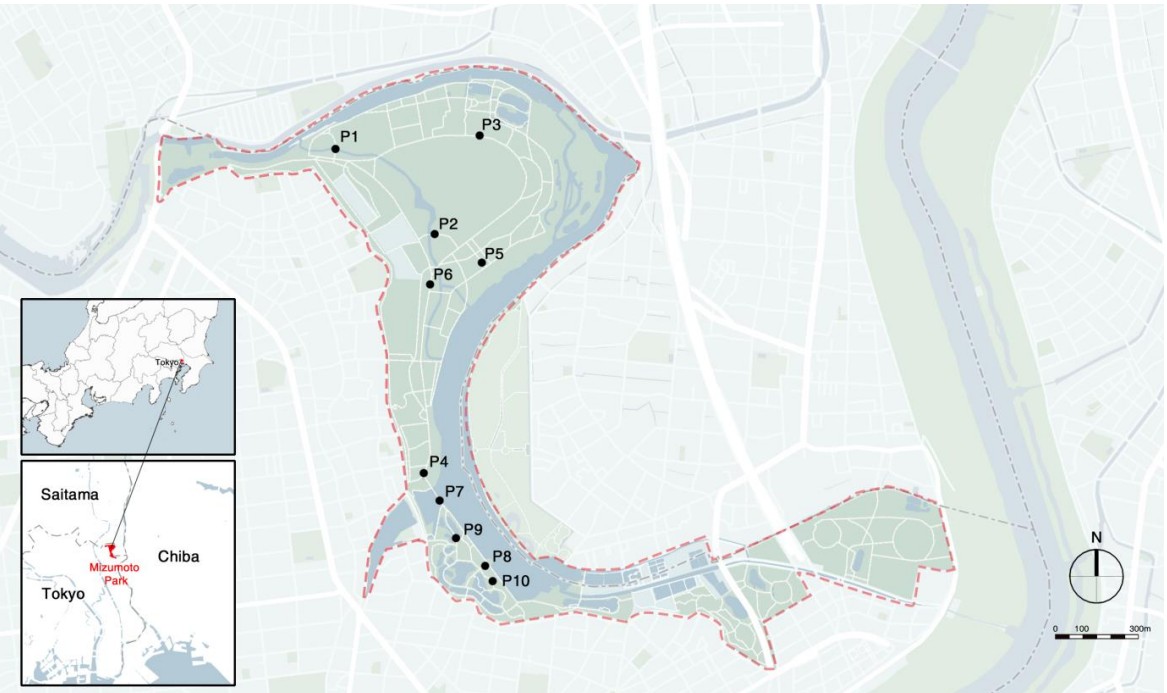

**Figure 1.** The scope of Mizumoto Park and photo locations.

## 2.2. Data Collection

Still images can capture many important properties of natural scenes [39]. In the visual aspect of landscape studies, photographs are considered valid surrogates for real scenes [40]. We chose a sunny day in early summer for the on-site photo shoot because visitors often participate in park activities at such a time. The researchers walked along the park road within the study site and took photos in all directions on 11 June 2021, between 2:00 and 4:00 p.m. We used a Canon 6D camera and Canon EF50 mm F1.4 USM lens. We divided the area into 500 m sections and collected more than 20 photos per section. The shooting height was 1.7 m, and the depth of field was the same. People were not deliberately avoided when capturing a photo. A total of 180 photos were obtained. Blurry and poorly lit photos were removed. In this study, we aimed to select some representative scenes in the park to explore people's viewing patterns and the influence of elements on viewing. Therefore, the remaining photos were screened based on the following principles: (1) the photos were divided into several categories according to the main landscape features, and two photos of different locations under each category were required; (2) the photos had to contain a variety of landscape elements, including natural elements and artificial elements; and (3) the subjects in the photos were popular attractions in the park (based on the park description and images on the park website and social media). A total of ten photos in five dominant landscape categories were retained and not altered in any way [41]. P1 (Youth Campground Entrance) and P2 (Barbecue Shop) were important structures in the park; P3 and P4 were the main squares; P5 and P6 were metasequoia and poplar woods, respectively; P7 and P8 were important waterside facilities for fishing and wild bird observation, respectively; and P9 and P10 were famous iris field landscapes (Figure 2). The shooting locations of the ten photos are marked in Figure 1.

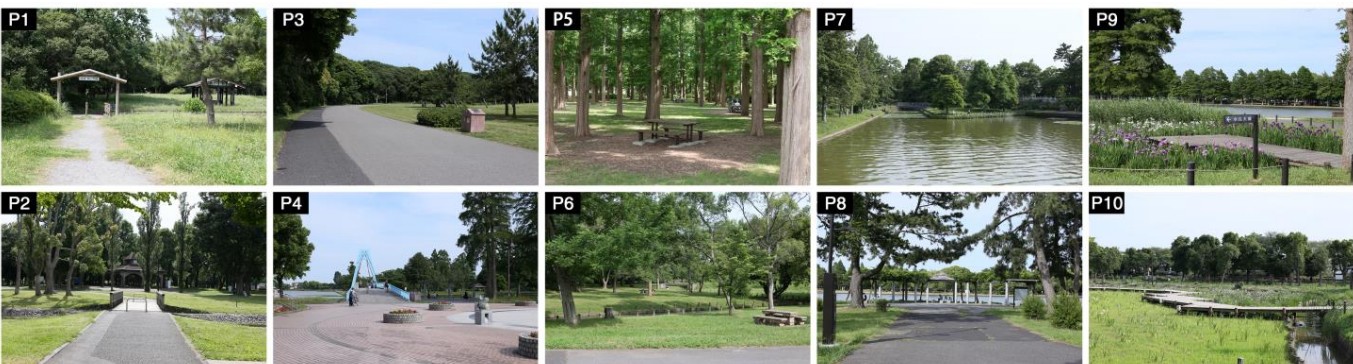

**Figure 2.** Ten selected photos.

We conducted an eye-tracking experiment in July 2021. The participants were residents living within 2 km of Mizumoto Park. To reduce the influence of other demographic variables (e.g., age, occupation, and education), participants were all current master's students at Chiba University Matsudo campus, aged between 22 and 28 years. A total of 16 students (eight men and eight women) who lived near the park and had normal eyesight and color vision were recruited through the Internet. All participants gave their written, informed consent to participate in the study. To avoid interference from sound and light, a dark and quiet room was used for the experiments. After a calibration procedure for the eye tracker, a slideshow of the photos was played for 10 s per slide in random order on a monitor with a resolution of $1920 \times 1080$. Participants were asked to observe the monitor. Tobii Pro Nano was used to record the eye-tracking data of each participant. Gaze heat maps, gaze plot maps, and related time data were exported using Tobii Pro Lab Screen Edition. After the eye-tracking experiment, we asked the participants to rate the visual aesthetic quality (VAQ) of the photos on a five-point linear scale with textual description endpoints ranging from "very ugly" (1) to "very beautiful" (5) [42]. This study was conducted in accordance within the guidelines of the Declaration of Helsinki. The study was noninvasive and did not investigate human or physiological data; all data were processed in an anonymized form. According to the institutional guidelines of Chiba University, there was no need to submit material for ethical review.

*2.3. Data Analysis*

Commonly used eye-movement information provided by eye trackers mainly includes fixation, saccade, and follow [26]. In eye-tracking experiments using static images as materials, the eye-movement behaviors involved are predominantly fixation and saccade [33]. Referring to the study of Dupont, Antrop, and Van Eetvelde (2014) [43], a fixed position of the eyes for at least 100ms was considered fixation; a saccade was defined as an eye movement that moved the eyes to the next viewing position. Four major indicators were calculated: fixation count (FC), indicating the total number of fixation points in a photo; fixation duration average (FDA) [44], indicating the average time spent on each fixation; saccade count (SC), indicating the number of saccade behaviors per photo; and scan path length (SPL), indicating the total path length of the eye moving on the photo within 10 s of the photo being displayed [40].

During the visual experience, participants adjusted their eye movements to accommodate the demands of attention. Their eyes focused on areas of information or interest [39]. The gaze heat map presents the cumulative results of the participants' attention centers, with red indicating longer total fixation at a particular position. The heat map revealed that there were no gender differences regarding the most interesting object in each photo. Therefore, we marked the visual focus area (VFA) of each photo based on the cumulative heat map of all participants (Figure 3). Time to first fixation (TTFF) and fixation duration (FD) on VFA were calculated. TTFF is the amount of time that it takes a respondent to look at the area, starting from the moment the photo appears. FD is the sum of fixation times on this area [28].

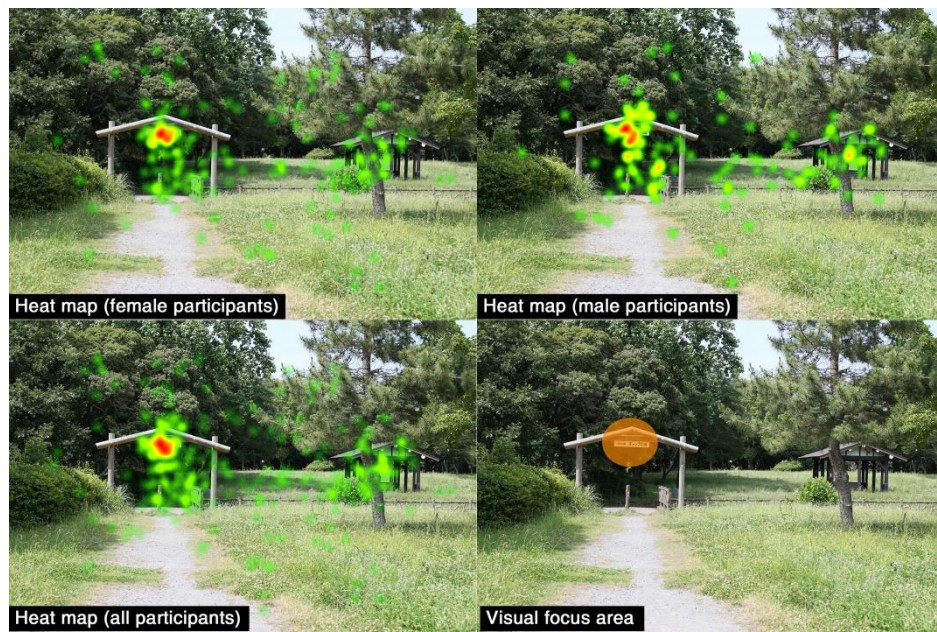

**Figure 3.** Example of gaze heat map and visual focus area (VFA).

This study divided the elements within photos into seven categories: sky, trees, lawn, other plants, paths/grounds, a water body, and artificial elements (Figure 4). Other plants refers to plants other than trees and grasses, such as shrubs, flowers, aquatic plants, and climbing plants. Paths/grounds refers to the park path and square grounds not covered by grass. Artificial elements refers to park facilities, structures, and visitors. The proportions of each category were determined in Adobe Photoshop CC. FD in each element's area was calculated. T-tests of eye-movement indicators and their Pearson correlations for elemental proportions were conducted using IBM SPSS Statistics 26. Then, *t*-tests of elemental gaze distributions and their correlation with VAQ were conducted.

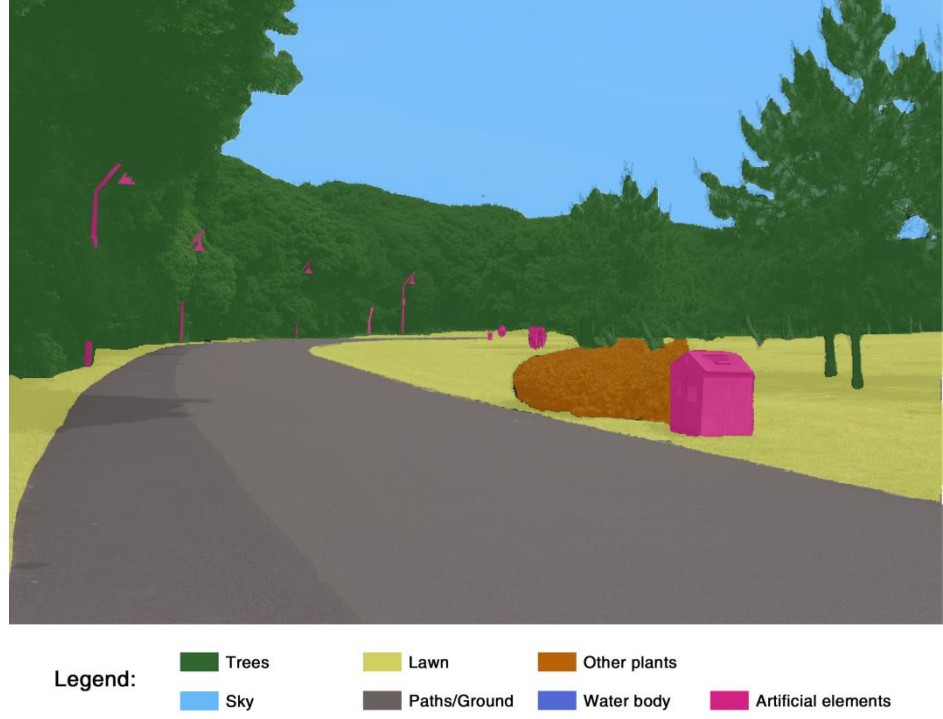

**Figure 4.** Example of element area division.

## 3. Results

### 3.1. Gender Differences in Eye-Movement Patterns

Figure 5 presents the mean results of the four eye-movement indicators for the two groups and the significant differences on the four items. Men exhibited more FC, SC, and longer SPL than women, whereas women demonstrated longer FDA. Figure 6 shows the gaze plot for P5 and P6 as representative samples, which displays the observation trajectories of the participants. The colored circles represent gaze points, circles size shows the duration of fixation, and numbers represent the fixation sequence. The figure intuitively reflected that within the same scene, gaze points were more scattered and distributed within a larger range among men than women.

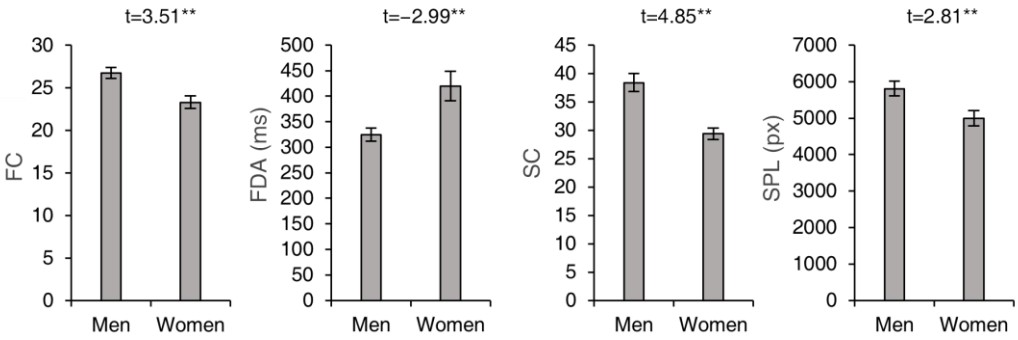

**Figure 5.** Independent sample *t*-test results of eye-movement indicators on the overall scene. Note: Bars represent the standard error of the mean; ** $p < 0.01$. FC = fixation count; FDA = fixation duration average; SC = saccade count; SPL = scan path length.

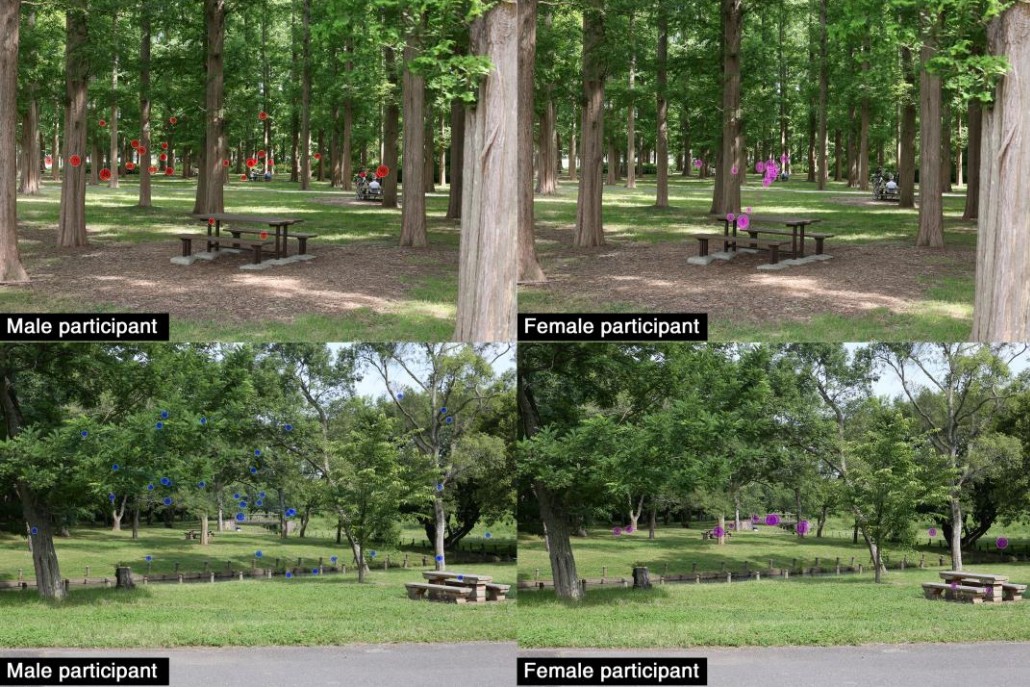

**Figure 6.** Gaze plots for representative samples.

The heat maps indicated that VFAs were generally located near the center of the images. Main objects in VFAs were visitors, structures, park facilities, and solitary shrubs. Previous studies have revealed that such elements have strong visual appeal [28,45]. Similarly, these elements attracted the attention of both genders in this study. However, the way participants focused on VFA differed. As shown in Figure 7, women had a longer FD. On

TTFF, there was a more significant difference between genders. Specifically, after switching to a new photo, women's gaze quickly (within 1 s) moved to the VFA near the center of the screen. In contrast, men scanned the scene extensively before entering the VFA, although they subsequently fixated on this area several times. Moreover, the resulting dispersion of TTFF for men was relatively large. There were large differences between individuals or for the immediate attractiveness of VFA for men.

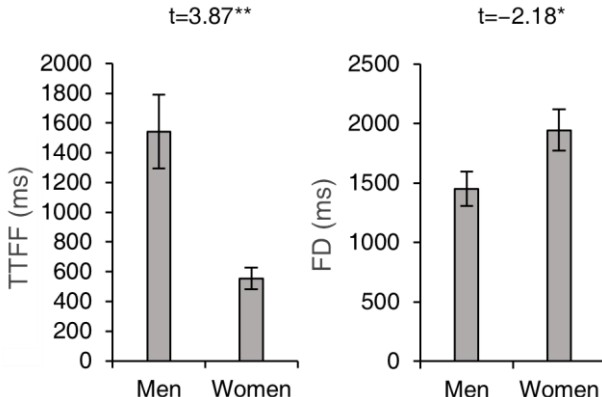

**Figure 7.** Independent-sample *t*-test results for eye-movement indicators on the visual focal area (VFA). Note: Bars represent the standard error of the mean; * $p < 0.05$, ** $p < 0.01$. TTFF = time to first fixation; FD = fixation duration.

Table 1 presents the association between eye-movement patterns and element proportion by gender. For men, only TTFF on VFA showed a significant correlation with the artificial elements (sig. = 0.0479). Women's TTFF showed a stronger correlation with the proportion of artificial elements (sig. = 0.0003). In addition, the proportions of the sky, trees, and lawn had a more significant impact on women's visual behavior. A larger sky area resulted in more fixation points, longer eye movement distance, and shorter dwell time on each fixation point and VFA. The areas of trees and lawn had the opposite effect to the sky.

**Table 1.** Pearson's correlation for eye movement indicators and element proportion.

|  |  | Sky (N = 80) | Trees (N = 80) | Lawn (N = 80) | Other Plants (N = 64) | Paths/Grounds (N = 80) | Water Body (N = 48) | Artificial Elements (N = 80) |
|---|---|---|---|---|---|---|---|---|
| Men | FC | 0.071 | −0.065 | −0.016 | 0.080 | −0.141 | 0.049 | 0.078 |
|  | FDA | −0.099 | 0.059 | 0.096 | −0.028 | 0.069 | −0.059 | −0.092 |
|  | SC | 0.053 | −0.055 | 0.044 | 0.032 | −0.121 | −0.027 | 0.160 |
|  | SPL | 0.168 | −0.179 | −0.093 | 0.034 | −0.054 | 0.053 | 0.119 |
|  | TTFF on VFA | 0.217 | −0.215 | −0.004 | −0.190 | 0.079 | −0.042 | 0.232 * |
|  | FD on VFA | −0.212 | 0.130 | 0.187 | 0.070 | −0.051 | −0.026 | −0.136 |
| Women | FC | 0.268 * | −0.229 * | −0.207 | 0.116 | 0.071 | −0.079 | 0.147 |
|  | FDA | −0.232 * | 0.149 | 0.242 * | −0.029 | −0.072 | 0.024 | −0.072 |
|  | SC | 0.174 | −0.120 | −0.149 | 0.080 | 0.035 | −0.036 | 0.058 |
|  | SPL | 0.237 * | −0.201 | −0.137 | 0.061 | 0.028 | −0.066 | 0.180 |
|  | TTFF on VFA | 0.146 | −0.223 | 0.126 | −0.075 | −0.018 | −0.147 | 0.404 ** |
|  | FD on VFA | −0.367 ** | 0.261 * | 0.311 ** | −0.038 | 0.004 | −0.136 | −0.144 |

Note: * $p < 0.05$, ** $p < 0.01$. FC = fixation count; FDA = fixation duration average; SC = saccade count; SPL = scan path length; TTFF = time to first fixation; FD = fixation duration; VFA = visual focus area.

### 3.2. Gender Differences in Gaze Distribution for Elements

There were significant gender differences in attention distribution. As shown in Figure 8, the FD of men on trees was significantly higher than that of women, whereas

the FD of women on other plants and artificial elements was significantly higher than that of men. In the sample in Figure 6, women's eyes were focused on the artificial elements in the image and were confined to below the lower edge of the canopy. Conversely, men repeatedly scanned the branches and leaves. The heat map indicated that, among other plants, women focused more on flowers, including herbaceous flowers and flowering shrubs, than men. Furthermore, solitary shrubs were more attractive to women.

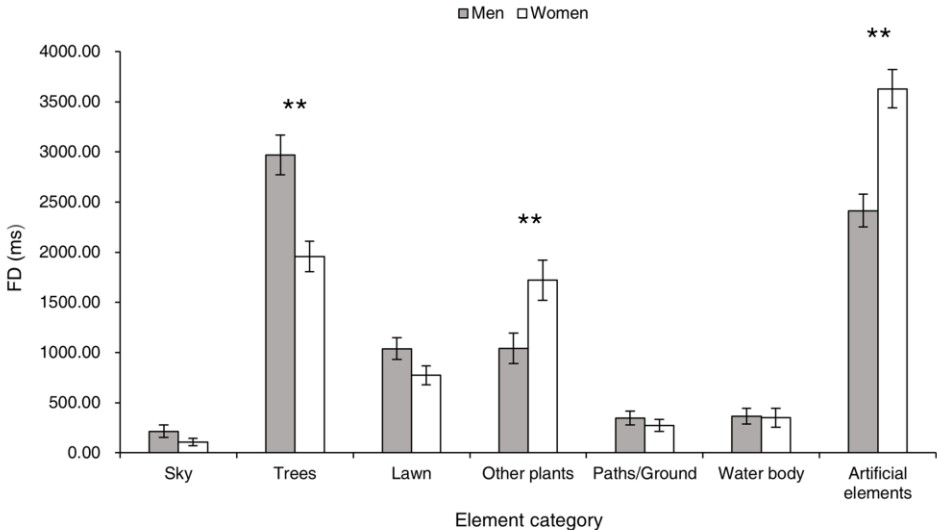

**Figure 8.** Element fixation duration (FD) mean scores. Note: Bars represent the standard error of the mean; ** $p < 0.01$.

A prolonged fixation on an element does not necessarily imply a strong liking or a higher rating [46,47]. Therefore, we explored the relationship between the aesthetic evaluation of the overall scene and gaze distribution on elements (Table 2). The positive correlation between the VAQ for scene and the FD for other plants was common between men and women. In contrast, sky showed a negative correlation, which was more pronounced among men. Moreover, VAQ for scene and the FD for paths/grounds demonstrated a significant negative correlation among women but not men.

**Table 2.** Pearson's correlations of element fixation duration (FD) and visual aesthetic quality (VAQ).

|  | Sky (N = 80) | Trees (N = 80) | Lawn (N = 80) | Other Plants (N = 64) | Paths/Grounds (N = 80) | Water Body (N = 48) | Artificial Elements (N = 80) |
|---|---|---|---|---|---|---|---|
| Men | −0.309 ** | −0.090 | 0.008 | 0.325 ** | −0.110 | 0.178 | 0.045 |
| Women | −0.270 * | −0.184 | 0.131 | 0.402 ** | −0.373 ** | 0.213 | −0.173 |

Note: * $p < 0.05$, ** $p < 0.01$.

## 4. Discussion

### 4.1. Gender Differences in Visual Behaviors during Landscape Viewing

In this experiment, the results revealed that male and female participants exhibited significantly different visual behaviors when viewing representative scenes in an urban park. Men moved their eyes more and observed a wider area, whereas women held their gaze at a fixation point for longer periods. Women were directly attracted to the VFA, whereas men often scanned other areas before entering the VFA. This noted difference in eye-movement patterns is relatively new to the field of landscape perception but fits with explanations for similar differences in other fields. Schulte, Hawelka, and Pletzer (2020) [48] were the first to discover the effects of sex hormones on visual behavior in number-comparison eye-tracking experiments. They found that estradiol and progesterone

were associated with more saccades and shorter fixations in men but not in women. Although the group of participants in their experiment differed from this study in some demographic characteristics (religion, culture, and occupation), this finding suggests that it is necessary to further explore the influence of sex hormones on landscape viewing in future research. Furthermore, FDA determined the ease with which scene information was extracted and encoded, with longer FDA reflecting greater effort by participants toward the objects being viewed [43,49]. Higher SC indicated that participants were searching or exploring more features in the scene [43]. Some studies have shown that men have an advantage in deciphering and remembering spaces. When exposed to new environments, men visually explore more spaces than women to understand the overall character and layout of the space [50]. Women generally rely more heavily on landmarks and exhibit sustained landmark-directed gaze during eye tracking, whereas landmark-directed gaze decreases over time among men [51]. In this study, facilities, structures, and solitary shrubs in the VFA served as landmarks, which could be the reason that the momentary attraction of VFA for women was stronger. The color or material of these iconic objects contrasted sharply with the environment, and women spent more time extracting information from them.

Women's eye-movement behavior was more influenced by element ratios than that of men. In our photo samples, the proportions of sky and trees were inversely related; thus, they had the opposite effect on the FC of women. Women visually explored open spaces with fewer trees more than enclosed spaces. Previous research reveals that women are more likely to perceive dense woodland as threatening [11,34]. Kaplan, Kaplan, and Ryan (1998) [52] propose that areas of blocked or obstructed visual pathways create fear. Tall, dense vegetation can obstruct one's view and provide concealment for perpetrators, and women are more sensitive than men to fear-inducing spatial conditions [53]. In this study, this sensitivity was reflected in the fact that women's sightlines rarely reached the dense canopy but were confined to its lower edge (Figure 6). This finding led us to consider that women's visual behavior may be related to safety perception. Furthermore, the significant positive correlation between FD on VFA and lawn ratio indicated that women focused more on iconic objects in scenes with larger lawn areas. There was no such phenomenon observed among men. This might be because large lawns appear homogeneous, requiring less information processing and allowing female observers to focus on heterogeneous parts of the image [54]. Although this study found a weak relationship between element ratios and men's eye-movement behavior, this does not necessarily mean that the proportion of landscape elements does not affect men's visual behavior.

Furthermore, women paid more attention to plants than men, especially flowers and shrubs. Previous studies reveal that women's preferences and attention to flowers are higher than that of men, which may be related to biological gender differences. For instance, the structure of the visual-processing pathway in women makes them more sensitive to the color of flowers [19,55]. Some studies showed that park users, especially women, prefer moderate vegetation density. The closer the woody plants are spaced, the less attractive they are [12,53]. The findings of this study are consistent with these notions. Shrubs in the photo samples were mostly dwarf shrubs with low density, which did not block the line of sight and were decorative; therefore, they were visually attractive. In addition, women also paid more attention to artificial elements than men. As mentioned, women rely on artificial elements to understand space. In addition to the heterogeneity of these elements within the scene [54], we speculate that this may be related to security perception. Based on previous literature, buildings are subconsciously defined as shelters that protect people from potential dangers [56]. Signage and low open-type fencing symbolize the control and guidance of the park on the movement routes of visitors [57,58]. Therefore, scenes with these elements could make women feel safer than completely natural scenes. Moreover, the presence of people in a park increases the feeling of safety [35,59]. Visitors walking or resting, as autonomous elements in the park, had strong visual appeal, and women spent more time than men confirming visitors' status and behavior.

Exploring the relationship between the FD on elements and VAQ of the scene revealed that men and women who gazed at shrubs and flowers for a longer time gave higher evaluations. Therefore, the eye-catching quality of these elements had a positive effect on the environment. Fixation on the sky was negatively correlated with photo scores. As sky is similar across photos in the same weather, looking at the sky longer could mean that the landscape beneath it appeared less interesting. This phenomenon was more pronounced in men, who provided lower ratings when looking at the sky for longer time. Furthermore, gaze on paths/grounds among women led to lower scores. Hard road surfaces with irregular paving or unevenness were more conspicuous. From an aesthetic point of view, this attention-grabbing had a negative effect.

### 4.2. Improvement Suggestions Based on the Female Perspective

Urban park design generally starts from a gender-neutral perspective, ignoring differences in needs depending on gender, particularly women's sensitivity to the environment. Some gender-sensitive design patterns attempt to divide physical space to accommodate different groups, which may exacerbate gender segregation. This study considered the visual perception to improve the inclusiveness of public space. According to the experimental results and related literature, it is concluded that women are more sensitive than men to perceiving danger and chaos. Enhancing the visual sense of security and order of the landscape could improve women's experience without damaging men's experience, allowing all visitors to experience fair treatment. Based on experimental results, we propose the following suggestions:

First, the vegetation density should be moderate and not obstruct the view. For trees, lower branches should be removed, and the canopy should be kept at eye level. Shrubs should not be too tall and completely enclosed. Women's gaze is more likely to explore and diverge in space with few trees and transparent light. In spaces surrounded by greenery, especially those with high lawn coverage, women tend to focus on key landscapes. Space planners and managers should make rational use of these two features based on design purposes. In addition, from the perspective of aesthetic preferences, plants such as decorative shrubs and flowers have a positive effect on the environment and are appreciated by viewers regardless of gender. Therefore, they need to be properly planted and maintained.

Second, visual focal points are important for women and help them understand the space. The design and management of landscapes should include visual focus. For instance, structures on the edge of water and woods, signs on the side of roads, and facilities for people to rest in the woods could aesthetically prevent the homogeneity of the landscape, and make women feel safe. Furthermore, the exterior design should be integrated into the surrounding environment.

Third, attention-grabbing park roads reduce aesthetic evaluation for women. The hard pavement should be smooth, and the coordination of color and pattern should be ensured when repairing. Nature trails require regular weeding [60]. In addition, routes with balanced user density make women feel more at ease; therefore, it is necessary to ensure that a certain number of people can be kept in the sight of others through the road design and management [1].

However, it is worth noting that the participants in this study are limited to some young people who are urban residents in Tokyo, Japan, and the results presented may not be universal worldwide. For example, although East Asian countries have reached more egalitarian gender relations in recent decades, they have maintained a more traditional gender climate as compared to Western societies, which may result in different visual behavior [61]. Nonetheless, the methodology (eye-tracking combined with subjective evaluation and elemental decomposition) and indicators used in this study can provide a theoretical framework for research and education to investigate group differences in landscape perception, including but not limited to other landscape types, other geographic areas, and other age groups.

### 5. Conclusions

This study explored gender differences in viewing urban park landscapes by tracking objective visual behavior and identified visual perception as one of the factors that contribute to the different experiences of men and women. Using data analysis and visualization, this study demonstrated that men have more saccades and a wider range of observations, whereas women spend more time gazing and pay more attention to iconic elements. In addition, women's eye movement behavior is more influenced by elemental ratios than males. Moreover, this study found that men gazed longer at trees and shorter at shrubs, flowers, and artificial elements than women. Attention-grabbing paths/grounds had a negative effect on the aesthetic evaluation of the scene among women but not men. These findings imply that women's visual experience of landscapes was more likely to be influenced by the safety and order provided by green spaces than men's. Based on these findings, suggestions for optimizing women's experiences at the visual level are proposed. Some examples are pruning low branches and high shrubs periodically to reduce blind spots, consciously adding visual focal points in the design process to help women comprehend the space, and reducing unsafe trails and ensuring smooth pavements. This study can inform designers and managers in creating targeted, inclusive green spaces that allow different user groups to benefit equally from urban public green spaces. This study's methodology may provide a reference point for other studies exploring group differences in landscape perception.

This study has some limitations. Firstly, as this study mainly focused on differences between genders, only participants with similar age and education levels were selected. As other demographic characteristics were not examined, the participants in this study may not be representative of the general population. The present research methodology should be replicated with different populations. In addition, based on such restrictive participant selection criteria, the sample size of recruited participants was relatively small which also led to weak representativeness of the results. Therefore, this study serves only as a methodological exploration within this field. On this basis, further studies can be carried out to extend the scope of the findings and verify the feasibility of the methodology. Secondly, to avoid high blink rates and loss of eye-tracking accuracy resulting from participant fatigue, the number of photo samples selected in this study was limited, resulting in weakened significance of the results. Thirdly, the static image with a fixed viewing angle cannot fully reflect the scene in the real environment which could have led to deviations in the results. Future studies should consider replicating the present experiment in virtual reality or real environments.

**Author Contributions:** Conceptualization, R.M.; methodology, R.M., Y.L. and K.F.; software, R.M.; formal analysis, R.M.; investigation, R.M. and Y.L.; data curation, R.M.; writing—original draft preparation, R.M.; writing—review and editing, K.F.; visualization, R.M.; supervision, K.F.; project administration, K.F. All authors have read and agreed to the published version of the manuscript.

**Funding:** This research received no external funding.

**Institutional Review Board Statement:** This study was conducted in accordance within the guidelines of the Declaration of Helsinki. The study was noninvasive and did not investigate human or physiological data; all data were processed in an anonymized form. According to the institutional guidelines of Chiba University, there was no need to submit material for ethical review.

**Informed Consent Statement:** Informed consent was obtained from all subjects involved in the study.

**Data Availability Statement:** Not applicable.

**Acknowledgments:** We thank all reviewers for their valuable comments on this paper.

**Conflicts of Interest:** The authors declare no conflict of interest.

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
