# Peer review of "Gender Differences and Optimizing Women’s Experiences: An Exploratory Study of Visual Behavior While Viewing Urban Park Landscapes in Tokyo, Japan"

_sustainability, doi:10.3390/su15053957_

Round 1
Reviewer 1 Report
The theme of the paper is interesting and original. The organization of the research is rather clear and rigorous, and well explained.
Like all research carried out on a statistical basis, there are some limitations. A certain limit is the size of the analysis sample (perception is a cultural, geographical and anthropological datum, as well as a physical one) which limits its application.
Similarly, some automatisms can be found in the "discussion", where mechanisms to increase the quality of perception are proposed in a simplified way. On the contrary, the results could (or should) be considered in an open form, as a further useful element for the complexity of the design processes.
In any case, this experience can be significant in defining a broad field of investigation, which links design to other scientific disciplines.
Author Response
We would like to thank you for your careful and thorough reading of this manuscript and for the thoughtful comments and constructive suggestions, which help to improve the quality of this manuscript. Here is a point-by-point response to your comments and concerns. All page and line numbers refer to the revised manuscript file.
Point 1:
Like all research carried out on a statistical basis, there are some limitations. A certain limit is the size of the analysis sample (perception is a cultural, geographical and anthropological datum, as well as a physical one) which limits its application.
Response 1:
Thank you for pointing this out. based on your suggestion, we highlighted limitations imposed by sample size in the “Discussion” and “Conclusions".
Details are as follows:
(Line 386-388) However, it is worth noting that the participants in this study are limited to some young people who are urban residents in Tokyo…
(Line 420-422) In addition, based on such participant selection criteria, the sample size of participants we recruited was relatively small, which also led to weak representativeness of the results.
Point 2:
Similarly, some automatisms can be found in the "discussion", where mechanisms to increase the quality of perception are proposed in a simplified way. On the contrary, the results could (or should) be considered in an open form, as a further useful element for the complexity of the design processes.
Response 2:
Thank you for this thoughtful insight. Based on this comment, we have reflected that other demographic characteristics such as culture and geography limit the generality of the differences proposed in this study. Therefore, we added some descriptions of limitations in the “Discussion” to narrow the reference range of the research conclusion itself but emphasized the reference value of this study as a theoretical framework.
Details are as follows:
(Line 288-290) Although the group of participants in their experiment differed from this experiment in some demographic characteristics (religion, culture and occupation) …
(Line 386-395) However, it is worth noting that the participants in this study are limited to some young people who are urban residents in Tokyo, Japan, and the results presented may not be universal worldwide….
Reviewer 2 Report
This research article has demonstrated interesting insights into the gender difference of preference in the park landscape characteristics using eye tracking technology. The results drawn from the eye-tracking experiment have provided evidence of the general preferences and contributed to informing the future design of the park landscape.
However, I believe the selection of the photos to conduct the experiment should be more systematically selected, such as photos selected deliberately having a contrasting element ( photos with only tall trees vs photos only with lower shrubs and photos with combinations of 2, etc). A more clear narrative of what the "landscape element" is tested may increase the robustness of the result. Secondly, the number of participants is relatively small. It is crucial to list this "small sample" as a limitation.
Author Response
We would like to thank you for your careful and thorough reading of this manuscript and for the thoughtful comments and constructive suggestions, which help to improve the quality of this manuscript. Here is a point-by-point response to your comments and concerns. All page and line numbers refer to the revised manuscript file.
Point 1:
I believe the selection of the photos to conduct the experiment should be more systematically selected, such as photos selected deliberately having a contrasting element ( photos with only tall trees vs photos only with lower shrubs and photos with combinations of 2, etc). A more clear narrative of what the "landscape element" is tested may increase the robustness of the result.
Response 1:
Thank you for this thoughtful insight. In the experiment, we selected photos according to the scene, mainly considering the element richness, popularity, function, and location of the scene, and then use the element type and proportion in the scene as indicators. Our original intention was to understand how users actively select objects to observe among many elements in a representative scene of the park. To express this, we have added something to the paper. It is true that current selection methods lack systematicity in defining "landscape element". And your suggestions about comparisons and combinations were very helpful. But in the case that the experiment has been completed, we are sorry that it is difficult for us to select pictures in a new way. We would like to refer to your suggestions in the future research to improve the complexity and rigor of the experimental design.
Details are as follows:
(Line 70-71) And the type and proportion of these landscape elements may affect their evaluation of the scene.
(Line 128-130) In this study, we aimed to select some representative scenes in the park to explore people's viewing patterns and the influence of elements on viewing.
Point 2:
Secondly, the number of participants is relatively small. It is crucial to list this "small sample" as a limitation.
Response 2:
Thank you for pointing this out. based on your suggestion, we highlighted limitations imposed by sample size in the “Discussion” and “Conclusions".
Details are as follows:
(Line 386-388) However, it is worth noting that the participants in this study are limited to some young people who are urban residents in Tokyo…
(Line 420-422) In addition, based on such participant selection criteria, the sample size of participants we recruited was relatively small, which also led to weak representativeness of the results.
Reviewer 3 Report
Abstract
- Show important statistical values that show differences in study results to increase comprehension and interest in the article.
Introduction
- Check additional documents on the use of green space How important it is to users to increase the interest of this research study.
Materials and Methods
- 2.1. Study site. Some explanations should be added to the introduction section. This section shows only the area (map) and what is used as educational equipment.
- 2.2. Data collection
Choosing the shooting time by reasoning that when the weather was sunny should be a time rationale that people come to do activities in the park instead.
3. Results
- Include statistical figures to explain study results in every section of study results.
4. Discussion
- Added a discussion of the effects of the use of research tools applied to research of the same nature as additional
5. Conclusions
- Increasing the experience of women to be more detailed, how to adjust, how to increase or decrease, in what direction, etc. to correspond with the title on the topic Optimizing the experience of women in this article.
Author Response
We would like to thank you for your careful and thorough reading of this manuscript and for the thoughtful comments and constructive suggestions, which help to improve the quality of this manuscript. Here is a point-by-point response to your comments and concerns. All page and line numbers refer to the revised manuscript file.
Point 1:
Abstract
- Show important statistical values that show differences in study results to increase comprehension and interest in the article.
Response 1:
Thank you for this thoughtful insight. But unfortunately, due to the word limit of the abstract, it is difficult for us to show important statistical values in the abstract without affecting the expression of background, methods, results, and conclusions.
Point 2:
Introduction
- Check additional documents on the use of green space How important it is to users to increase the interest of this research study.
Response 2:
Thank you for pointing this out. We added text about the importance of urban green space and users’ needs.
Details are as follows:
(Line 27-31) Using urban green spaces represented by urban parks can improve the physical and mental health of urban residents, as well as relieve anxiety and stress…
(Line 40-44) Previous research suggests perceived quality is a significant predictor of green space visit frequency…
Point 3:
Materials and Methods
- 2.1. Study site. Some explanations should be added to the introduction section. This section shows only the area (map) and what is used as educational equipment.
Response 3:
Thanks for your comment. We have moved some explanations from the “Study site” to the “Introduction”.
Details are as follows:
(Line 82-89) Therefore, based on the two studies, we conducted such a study in Japan. Several studies have reported that women have ambivalent feelings…
Point 4:
- 2.2. Data collection
Choosing the shooting time by reasoning that when the weather was sunny should be a time rationale that people come to do activities in the park instead.
Response 4:
Agree with this comment. Based on this comment, we have rewritten this content.
Details are as follows:
(Line 121-124) We chose a sunny day in early summer for the on-site photo shoot because visitors often do activities in the park then…
Point 5:
- Results
- Include statistical figures to explain study results in every section of study results.
Response 5:
Thanks for your comment. We have changed the independent sample t-test results (Table 1 and Table 2) in each section to be expressed in statistical figures (Figure 5 and Figure 7). But sorry, we found the Pearson correlation test results to be more readable in tables, so we kept tables.
Details are as follows:
(Line 213) Figure 5
(Line 235) Figure 7
Point 6:
- Discussion
- Added a discussion of the effects of the use of research tools applied to research of the same nature as additional
Response 6:
Thank you for this thoughtful insight. Based on your comment, we have added a discussion of the applicability of this research methodology.
Details are as follows:
(Line 386-395) However, it is worth noting that the participants in this study are limited to some young people who are urban residents in Tokyo, Japan, and the results presented may not be universal worldwide….
Point 7:
- Conclusions
- Increasing the experience of women to be more detailed, how to adjust, how to increase or decrease, in what direction, etc. to correspond with the title on the topic Optimizing the experience of women in this article.
Response 7:
Thank you for pointing this out. We have added detail to our suggestions for optimizing the experience for women in the conclusion.
Details are as follows:
(Line 410-413) For example, pruning low branches and high shrubs in time to reduce blind spots; consciously adding visual focal points in the design process to help women understand the space; reducing unsafe trails and ensuring smooth pavement.
Once again, we thank you for the time you put in reviewing our paper and look forward to meeting your expectations.
Round 2
Reviewer 3 Report
This article has been fully revised according to the instructions.
Author Response
Thank you for the time you put into reviewing our paper. And thanks for your suggestions, which help improve this manuscript's quality.